# Apples to Apples? A Comparison of Real-World Tolerability of Antiretrovirals in Patients with Human Immunodeficiency Virus Infection and Patients with Primary Biliary Cholangitis

**DOI:** 10.3390/v14030516

**Published:** 2022-03-03

**Authors:** Shannon L. Turvey, Lynora Saxinger, Andrew L. Mason

**Affiliations:** 1Division of Infectious Diseases, Faculty of Medicine and Dentistry, University of Alberta, Edmonton, AB T6G 2E1, Canada; sturvey@ualberta.ca (S.L.T.); saxinger@ualberta.ca (L.S.); 2Division of Gastroenterology, Faculty of Medicine and Dentistry, University of Alberta, Edmonton, AB T6G 2E1, Canada

**Keywords:** human immunodeficiency virus infection, combination antiretroviral therapy, human betaretrovirus, primary biliary cholangitis

## Abstract

We previously characterized a human betaretrovirus and linked infection with the development of primary biliary cholangitis (PBC). There are in vitro and in vivo data demonstrating that antiretroviral therapy used to treat human immunodeficiency virus (HIV) can be repurposed to treat betaretroviruses. As such, PBC patients have been treated with nucleoside/nucleotide reverse transcriptase inhibitors (NRTIs), alone and in combination with a boosted protease inhibitor or an integrase strand transfer inhibitor in case studies and clinical trials. However, a randomized controlled trial using combination antiretroviral therapy with lopinavir was terminated early because 70% of PBC patients discontinued therapy because of gastrointestinal side effects. In the open-label extension, patients tolerating combination therapy underwent a significant reduction in serum liver parameters, whereas those on NRTIs alone rebounded to baseline. Herein, we compare clinical experience in the experimental use of antiretroviral agents in patients with PBC with the broader experience of using these agents in people living with HIV infection. While the incidence of gastrointestinal side effects in the PBC population appears somewhat increased compared to those with HIV infection, the clinical improvement observed in patients with PBC suggests that further studies using the newer and better tolerated antiretroviral agents are warranted.

## 1. Introduction

### 1.1. Primary Biliary Cholangitis

Primary biliary cholangitis (PBC) is a complex autoimmune liver disease that predominantly occurs in women [1,2]. The disease is characterized by the production of anti-mitochondrial antibodies and immune-mediated destruction of interlobular bile ducts. PBC risk is elevated in related and unrelated family members, implicating genetic predisposition and environmental factors in the disease process [1,2,3]. Although autoimmunity is commonly thought to cause disease, studies using immunosuppression and biological therapy have been inconclusive, and individual treatments have not been adopted because of toxicity or a lack of efficacy [4,5].

There is no curative therapy, and there is a definitive need for better strategies to manage the symptoms of fatigue and itch that commonly occur in patients with PBC. Choleretic agents are used to improve bile flow and inflammation, but 50% of patients develop progressive ductopenia and cirrhosis [1,2]. Liver transplantation is the only therapeutic intervention for those who develop liver failure, and post-transplant, systemic symptoms such as chronic fatigue may persist in a large proportion of patients. Up to 50% of patients develop recurrent disease in the allograft, and the appearance of biochemical cholestasis soon after liver transplantation predicts the subsequent development of histological recurrence of PBC at a later date [6]. Early post-transplant disease recurrence is more suggestive of an infectious disease process in the allograft because an autoimmune attack is less likely shortly after transplantation when immunosuppression is highest [7]. Indeed, recurrent PBC after liver transplantation remains a common and clinically important issue, and the potential to treat an infectious cause of PBC remains a priority. It is therefore of interest that combination antiretroviral therapy (cART) has shown some utility in reversing cholangitis in patients with PBC who experience disease recurrence following liver transplantation (Figure 1) [8].

### 1.2. Human Betaretrovirus

The classification of betaretroviruses is somewhat confusing because of different nomenclature adopted for exogenous and endogenous viruses [10]. The Betaretrovirus genus comprises the mammalian exogenous and endogenous retroviruses, such as the mouse mammary tumor virus (MMTV) and the Mason-Pfizer monkey virus, which represent B-type and D-type retroviruses, respectively, as well as the HERV-K group of human endogenous retroviruses. Mammalian exogenous retroviruses and HERV-K human endogenous retroviruses tend to form separate branches of the betaretrovirus phylogenetic tree and are therefore more distantly related [11]. There exist over 1000 characterized endogenous HERV-K sequences within the human genome, with the most recent germline addition referred to as HML-2 elements (human endogenous MMTV-like 2) [10,11]. While these endogenous viruses may be active and, on rare occasions, form viral particles, none have been characterized as exogenous transmissible agents.

The human betaretrovirus (HBRV) is the only exogenous betaretrovirus characterized in humans to date, and the agent is not endogenously encoded in the human genome (although mistakenly reported as an endogenous retrovirus [12]). HBRV shares close genomic similarity with MMTV to the extent that the two viruses are difficult to distinguish genetically [13]. Therefore, HBRV is more closely related to mammalian exogenous retroviruses than human endogenous retroviruses, such as the HML-2 family [11]. MMTV is the causal agent of breast cancer in mice, and the original discovery of the mouse betaretrovirus was complicated by the observation that the agent could be passaged as an exogenous virus in milk and transmitted endogenously in the germline as an endogenous virus [3,14].

The human betaretrovirus was first characterized in patients with breast cancer over 50 years ago, when evidence of B-type particles resembling MMTV was detected in the milk of 60% of patients with breast cancer [15,16]. Although additional data supporting the presence of HBRV in breast cancer emerged throughout the 1970s, interest declined due to concerns of false reactivity to HERV-K that had been shown to be expressed in breast cancer tissues and the lack of reproducible assays to detect HBRV. In the 1990s, HBRV (then referred to as the human mammary tumor virus) was cloned from breast cancer samples. However, research was hindered by the lack of reproducible diagnostic assays, the low viral burden and concerns that PCR studies in patients may be confounded by contamination with mouse DNA [16,17,18]. Subsequently, MMTV-like sequences were cloned from patients with PBC, and the agent was referred to as HBRV, in keeping with the International Committee of Taxonomy of Viruses [13].

### 1.3. HBRV and PBC

The first indication that patients with PBC may have viral infections surfaced in a serological study reporting retroviral antibodies in patients with PBC and related idiopathic cholestatic disorders [19]. Electron microscopy studies revealed evidence of virus-like particles in biliary epithelial cells extracted from patients with PBC prior to cloning HBRV sequences, first from a biliary epithelial cell library and then perihepatic lymph nodes, a major reservoir for HBRV in patients with PBC [13,20]. Initially, the role of viral infection in PBC was questioned because the viral burden was insufficient to detect HBRV RNA in the liver of most PBC patients tested [20]. As a result, investigators suggested that viral integration studies be conducted to provide the “final evidence” for the role of the betaretrovirus in PBC. Accordingly, ligation-mediated PCR and next-generation sequencing were used to demonstrate over 1500 HBRV insertion sites in the cholangiocytes and lymph nodes of the majority of PBC patients tested [21].

ELISA studies using HBRV Env have revealed significantly increased seroprevalence in PBC patients and breast cancer patients as compared to age-matched controls [22]. However, the serological frequency was somewhat low, possibly related to the presence of immunosuppressive domains harbored by the betaretrovirus Env protein [23]; similar observations have been made in neonatal mice, in which MMTV infection triggers IL-10 production, limiting neutralizing antibody formation [24]. Nevertheless, a much higher prevalence of infection has been detected using cellular immune assays, and intrahepatic lymphocytes from PBC patients undergoing liver transplantation have been found to generate robust proinflammatory responses [25].

While these studies clearly placed HBRV infection at the site of disease, they did not provide evidence of transmissible infection. Accordingly, co-cultivation studies were performed to isolate transmissible HBRV from PBC lymph node homogenates with HS578T cells. Evidence for transmissible HBRV was demonstrated by electron microscopy, insertion sites in the newly infected human HS578T cells and passage of the virus to biliary epithelial cells [26]. To establish Koch’s postulates in vitro, co-culture studies were performed to show that purified viral particles could trigger a disease-specific phenotype of PBC in biliary epithelial cells associated with the autoimmune response [20,27]. Despite these studies implicating the virus in one aspect of the disease process, we currently lack methods to prove that an infectious agent may cause a complex multifactorial autoimmune disease influenced by genetic, environmental, and biological (sex) factors.

### 1.4. Proof-of-Principle ART Studies in PBC

Consequently, we previously conducted interventional studies with repurposed ART to address the hypothesis that HBRV infection is central to PBC and, at the same time, perform proof-of-principle studies to assess whether clinical improvement corresponds to a reduction in virologic load [8,28,29,30,31,32,33]. While the use of ART was associated with demonstrable biochemical and histological improvement, treatment with HIV protease inhibitors was associated with clinically significant side effects. Herein, we compare experiences in the experimental use of ART in patients with PBC and the broader use in people living with human immunodeficiency virus (HIV) infection (PLWH).

## 2. Antiretroviral Use and Adverse Effects in Primary Biliary Cholangitis

### 2.1. ART in PBC

Several ART agents used to treat PLWH have been repurposed for patients with PBC. Specifically, nucleoside/nucleotide reverse transcriptase inhibitors (NRTIs) have been tested in vitro, in animal models and in clinical trials with PBC patients, either alone or in combination with the protease inhibitor (PI) combination ritonavir-boosted lopinavir (LPV/r) or the integrase strand transfer inhibitor (INSTI) raltegravir (RAL) [8,28,29,30,31,32,33].

### 2.2. NRTI Use with Betaretrovirus and Development of Resistance-Associated Variants

Animal models have been used to test ART for activity against MMTV. Zidovudine (AZT) has demonstrable utility in inhibiting MMTV [34], and other ART agents have shown activity in reducing MMTV levels in the NOD.c3c4 autoimmune biliary disease mouse model of PBC [35]. However, use of AZT/lamivudine (3TC) without another agent in NOD.c3c4 resulted in virologic resistance with biochemical rebound and the emergence of unique variants in the MMTV polymerase (*pol*) gene [36]. Not surprisingly, similar observations were found using NRTI alone in patients with PBC, where biochemical breakthrough was observed with ART, although resistance-associated variants were not assessed in this study [31]. In open-label studies, PBC patients on twice-daily AZT/3TC experienced improved alkaline phosphatase (ALP) levels used to measure biliary damage and inflammatory disease on liver biopsy, whereas those on 3TC monotherapy did not experience any significant improvement [31]. Overall, treatment was well tolerated, but one patient stopped AZT/3TC due to nausea and fatigue [31].

Based on the biochemical improvement observed with 3TC/AZT, a 6-month multicenter randomized controlled trial was conducted in patients with PBC unresponsive to ursodeoxycholic acid, which is the standard of care [32]. Notably, patients developed a significant reduction in ALP as compared to those on placebo but failed to meet the stringent criteria for the study endpoints [32]. In total, two patients (7%) discontinued 3TC/AZT treatment because of anemia and fatigue. As previously observed in HIV clinical trials, it became apparent that the development of resistance to single or dual nucleoside therapy would necessitate the introduction of combination therapy with a third potent antiretroviral agent to suppress viral replication and reduce the risk of resistance [32].

### 2.3. Combination ART Use with Betaretrovirus

One of the first clinical reports of combination ART (cART) originated in a patient with HIV and HBRV coinfection who presented with biopsy-confirmed PBC and evidence of HBRV in whole blood by PCR [33]. The hepatic biochemistry improved considerably with the institution of a combination of tenofovir disoproxil fumarate (TDF) and emtricitabine (FTC) and the protease inhibitor (PI) lopinavir boosted with ritonavir (LPV/r). The patient’s liver tests then normalized completely once the choleretic ursodeoxycholic acid was commenced [33].

Subsequently, LPV inhibition of the betaretrovirus protease was confirmed in an in vitro assay [37] and then tested for in vivo activity using the NOD.c3c4 mouse model of MMTV cholangitis [36]. A comparative analysis of several dual NRTI regimens with and without LPV/r revealed that the combination of TDF/FTC with LPV/r had the optimal biochemical and histological impact and reduced MMTV RNA levels in the liver [36]. Then, a cART regimen with TDF/FTC and LPV/r was used to treat a young patient with severe recurrent PBC following liver transplantation [8]. Biochemical and histological improvements were observed, but LPV/r inhibited the metabolism of tacrolimus to such an extent that the patient only required 0.5 mg tacrolimus weekly to maintain adequate immunosuppression levels (Figure 1). The commencement of cART was also associated with biochemical hepatitis [8].

### 2.4. Randomized Controlled Trial of Combination TDF/FTC and LPV/r in Patients with PBC

A multicenter randomized controlled trial compared the same combination of TDF/FTC and LPV/r versus placebo in patients with PBC unresponsive to ursodeoxycholic acid, employing the same dosing regimen as that used for PLWH [28]. Initially, transient elevations in hepatic biochemistry occurred, and by the end of 6 months of treatment, a significant reduction in ALP from baseline was observed, accompanied by a reduction in HBRV DNA in whole blood [28]. However, the small number of patients treated and adherence challenges with LPV/r treatment prevent any firm conclusions from this study. Most patients on cART developed treatment-limiting side effects, and the study was terminated early with just 20% enrollment [28]. The majority of patients who tolerated therapy transitioned to the open-label/crossover phase and were treated for a total of 24 months with TDF/FTC and for variable lengths of time with LPV/r [29]. In total, 90% of PBC patients on treatment and 50% on placebo developed gastrointestinal (GI) side effects, including nausea, vomiting, diarrhea, constipation, weight loss, abdominal pain and bloating [28,29]. Including the open-label phase of the study, 70% discontinued LPV/r therapy due to GI side effects. Notably, only those that remained on treatment with TDF/FTC and LPV/r for the duration of the 2-year study developed maintained improvements in their liver tests (up to 40%), whereas those on TDF/FTC alone experienced biochemical rebound to baseline values [29]. PBC symptoms assessed by a validated scoring system [38] improved over the course of the 24-month study in patients on LPV/r. This is an important observation because no other therapies have been shown to impact cognition and fatigue [1,2].

## 3. Antiretroviral Adverse Effects in People Living with HIV (PLWH) Infection

### 3.1. Side Effects of ART

In 2020, UNAIDS reported that there were approximately 37.6 million people living with HIV infection (PLWH) globally, and this population requires ART to improve life expectancy [39]. However, the initial cART studies reported a plethora of adverse effects and toxicities [40]. An early review of outcomes in patients on one of six protease inhibitor or non-NRTI-based antiretroviral therapy regimens found that approximately 50% of patients discontinued the initial antiretroviral regimen within the first year of therapy [41]. When the reason for treatment discontinuation was captured, GI adverse events were the cause of discontinuation in 44% of cases [41]. In a recent cohort study of over 3000 PLWH starting ART, 12% of patients discontinued their initial regimen within the first year of therapy, with the highest discontinuation rates (29–35%) in patients using older, non-integrase strand transfer inhibitor (INSTI)-based regimens [42]. Intolerance or toxicity was the leading cause for stopping the initial ART regimen in this cohort and accounted for 45% of the discontinuation rate [42].

### 3.2. Side-Effect Profiles of Nucleoside/Nucleotide Reverse Transcriptase Inhibitors

NRTIs were the first class of antiretroviral drugs utilized clinically for HIV therapy and remain the backbone of ART in PLWH, in combination with a third agent from a different class [39]. Lipodystrophy and mitochondrial toxicity—including lactic acidosis, myopathy, neuropathy, and hepatotoxicity—were observed with older NRTIs that have since been largely abandoned [40,43]. Of the newer medications, abacavir (ABC), used in conjunction with 3TC, is associated with hypersensitivity reactions in patients positive for HLA-B5701 [44] and is possibly linked with elevated cardiovascular risk [45,46]. TDF, used in combination with FTC, is associated with proximal renal tubulopathy and Fanconi’s syndrome as well as with osteopenia and osteoporosis [43]. The newer formulation, tenofovir alafenamide (TAF), does not cause renal dysfunction but is associated with weight gain, an unwanted side effect in those with normal or elevated weight prior to starting therapy. Indeed, weight gain appears to be amplified with TAF in combination with second-generation integrase strand transfer inhibitors (INSTIs) such as dolutegravir or bictegravir [47,48,49,50]. Unlike some other antiretroviral drug classes, NRTIs are not specifically linked with diarrhea or other GI symptoms [51].

### 3.3. Side-Effect Profile of HIV Protease Inhibitors

PIs were the second class of antiretrovirals developed. While they saved many lives in the era of highly active antiretroviral therapy, their use was limited by GI toxicity and poor bioavailability with a need for pharmacokinetic boosting [52]. The most common adverse effects of PIs are weight gain, lipodystrophy and metabolic syndrome with dyslipidemia, insulin resistance and diabetes [52]. GI side effects, particularly diarrhea, were also commonly reported as a prominent issue impacting PI tolerability. The collective experience of using LPV/r to treat PLWH reveals that moderate to severe diarrhea occurs in approximately 20% of patients, with nausea occurring in 10% and vomiting occurring in 7% [53].

Patients treated with older PIs (saquinavir, indinavir, nelfinavir, amprenavir, fosamprenavir and tipranavir) experienced a comparable or higher frequency of side effects, and as such, these are no longer commonly used. For example, 21% of patients from a large cohort of treatment-naïve individuals who had started on PI-based regimens at the turn of the century eventually discontinued ART due to toxicity [54]. Nevertheless, boosted regimens incorporating atazanavir (ATZ) and darunavir (DRV) are still recommended as alternative first-line agents for patients with resistance to INSTI regimens [55]. LPV was the first second-generation PI that the FDA approved in 2000, and the treatment requires both BID dosing and boosting with ritonavir. LPV treatment is associated with metabolic syndrome, GI intolerance, Achilles tendinopathy and systemic hypersensitivity [52].

In total, discontinuation rates for LPV/r range from 3% to 19% in PLWH [56,57,58,59], whereas we found that 70% of patients with PBC terminated LPV/r treatment [28]. Although the limited experience of using LPV/r in PBC restricts the development of firm conclusions, the incidence of GI side effects appears to be double that observed in HIV. It is unknown why this is the case, but it is notable that patients with PBC have a higher prevalence of celiac and inflammatory bowel diseases in general [1,2]. In addition, it is important to recall that 50% of the patients on placebo in the cART study also reported GI side effects [28].

Hypercholesterolemia is also an issue with LPV/r use in PLWH and occurs in 7% of patients [53], whereas in PBC patients, hypercholesterolemia occurs as a result of cholestasis with the interruption of bile flow. However, this problem was only recorded in one PBC patient on placebo, who discontinued the study for this concern [28].

ATZ can be dosed once daily, and in contrast to older PIs, it is not associated with metabolic syndromes [52]. However, ATZ is associated with proximal tubulopathy and unconjugated hyperbilirubinemia, and absorption is impaired by higher gastric pH, such as with exogenous gastric acid suppression [52,60,61]. In a retrospective review of 108 patients on an antiretroviral regimen that included ATZ, 22% had clinically diagnosed jaundice [60]. In other studies, hyperbilirubinemia led to antiretroviral discontinuation in approximately 1% of patients on ATZ-based regimens [62,63,64].

In 2006, the FDA approved a once-daily dosed second-generation PI, DRV, which requires boosting with either ritonavir (DRV/r) or cobicistat (DRV/c) [52,65]. Initial studies showed that DRV-based regimens were associated with increased virologic suppression and a decreased risk of regimen discontinuation compared with other PI-based regimens [66]. Although DRV is linked with weight gain, diarrhea, and other GI side effects, these occur to a lesser degree as compared with LPV/r, with an overall 3% discontinuation in patients on DRV/r due to nonfatal adverse events [52,67,68]. Head-to-head studies of DRV/r versus LPV/r reported superior DRV/virologic suppression, with fewer adverse-event-related discontinuations (5% vs. 13%) and decreased treatment-related diarrhea (5% vs. 11%) [69].

### 3.4. Side Effects of Non-Nucleoside Reverse Transcriptase Inhibitors

Non-nucleoside reverse transcriptase inhibitors (NNRTIs) such as efavirenz, nevirapine and doravirine are associated with central nervous system side effects, systemic hypersensitivity reactions, hepatotoxicity, and dyslipidemia [40,51]. NNRTIs demonstrate superior tolerability as compared to PIs and integrase strand transfer inhibitors (INSTIs), with reduced GI adverse events such as diarrhea [51]. Nevertheless, diarrhea was the most common adverse event in the DRIVE-FORWARD study, which compared combination NRTIs with doravirine versus combination NRTIs and DRV/r [70]. The frequency of diarrhea was higher than that reported in prior DRV/r studies, where 13% of the patients in the DRV/r study arm experienced diarrhea versus 5% of those treated with doravirine [71]. To our knowledge, these agents have not been tested clinically for activity against HBRV [5].

### 3.5. Integrase Strand Transfer Inhibitors

INSTIs are now the first-line antiretroviral therapy for treatment-naïve PLWH [55]. The newer-generation INSTIs (dolutegravir and bictegravir) have a high genetic barrier to resistance and once-daily dosing. Dolutegravir and bictegravir are also both available in coformulations with NRTIs as single-tablet regimens for HIV treatment. While weight gain can occur after the initiation of any antiretroviral regimen and is at least in part associated with the “return to health” phenomenon, there is a clearly delineated association between the initiation of INSTI-based regimens—particularly dolutegravir and bictegravir—and weight gain compared with PI and NNRTI regimens [47,48,49,72,73,74,75]. Enhanced weight gain is a particular challenge in a population where diabetes, cardiovascular disease and other metabolic conditions are more prevalent than in the general population [72]. In contrast, an older INSTI, raltegravir, is not as clearly associated with weight gain [76]. However, raltegravir is often dosed twice daily and is not currently marketed as part of any single-tablet regimen for HIV treatment. Nevertheless, diarrhea continues to be an issue with INSTIs, as illustrated by the FLAMINGO study for treatment-naïve PLWH, where the frequency of DRV/r-related diarrhea was 29% as opposed to 17% of PLWH in the dolutegravir group [77].

### 3.6. Hepatotoxicity with Antiretroviral Drugs

Hepatotoxicity is a relevant consideration when looking at ART use in PBC, given that all classes of antiretroviral drugs used for HIV therapy have been shown to be associated with hepatotoxicity [78,79,80]. The reported incidence of hepatotoxicity after ART range from 8% to 24% in PLWH [80]. Up to 30% of patients experiencing hepatotoxicity may require a change in ART regimen [80]. There are four possible mechanisms of ART hepatoxicity, including direct drug hepatotoxicity, hepatic involvement in hypersensitivity reactions, mitochondrial toxicity, and immune reconstitution inflammatory syndrome [80,81]. Hepatotoxicity risk may be amplified in patients with viral coinfection, coexisting hepatic schistosomiasis or underlying liver disease [44,79,81,82].

Moderate to severe elevation in liver enzyme abnormalities are reported to occur in 3.5% of HIV patients receiving LPV/r [53]. In our PBC patients treated with LPV/r, a transient rise in aminotransferases was observed shortly after initiating treatment, which tended to resolve without the discontinuation of therapy [28]. While this may be related to transient LPV/r drug-induced liver injury, we characterized an immunosuppressive domain in the HBRV Env protein, suggesting that immune reconstitution with a reduction in viral load may provide an alternative hypothesis [23]. The propensity to develop chronic hepatitis was observed in a liver transplant recipient treated with LPV/r and TDF/FTC. In this instance, the decision to persist with cART was based on the repeated demonstration of histological improvement without histological evidence of parenchymal and interface hepatitis (liver biopsies 8 and 9 in Figure 1). Nevertheless, the consideration of changing to darunavir (DRV) was discussed based on the possible utility of this agent against HBRV. However, this was not necessary because the AST levels subsequently decreased towards baseline without any change in treatment (Figure 1).

## 4. Conclusions

The preliminary studies summarized here support both the potential role of HBRV in the development of primary biliary cholangitis and the continued investigation of cART in this challenging-to-treat patient group. Betaretroviruses appear to trigger the autoimmune phenotype, exposing mitochondrial antigens in vitro and in animal models in vivo. HBRV is found at the site of disease with proviral insertions in cholangiocytes, and PBC patients’ intrahepatic lymphocytes demonstrate cellular immune responses to HBRV [25]. Further, there is an evolving body of work demonstrating biochemical and histological responses to cART therapy in PBC patients. While clinical experience with combination ART is far greater in patients living with HIV than those with primary biliary cholangitis, known toxicities of cART appear to be exaggerated in patients with PBC, limiting the use of older agents. The prevalence of GI adverse events was substantially greater in patients with PBC than that reported in patients with HIV infection. This may be related either to the increased propensity for GI intolerance in PBC patients or to a lower threshold to report side effects based on treatment expectations. Of note, approximately 50% of patients with PBC do not have life-threatening disease, nor do they develop cirrhosis, and the usual PBC medications do not have a high burden of side effects (except for itching). Therefore, it is possible that PBC patients may be less willing to accept less tolerable regimens than HIV patients for whom cART is a lifesaving, lifelong therapy. Fortunately, the antiretroviral landscape continues to evolve better tolerated formulations. The investigation of cART in PBC allows both the further examination of disease pathogenesis and the therapeutic potential of regimens in this challenging disease. It will therefore be crucial to maximize the tolerability of cART regimens used in clinical trials for patients with PBC and compare side-effect profiles to the known tolerability data in PLWH. In this regard, another randomized controlled trial is now underway using combination TDF/FTC with RAL for patients with PBC (clinicaltrials.gov NCT03954327).

## Figures and Tables

**Figure 1 viruses-14-00516-f001:**
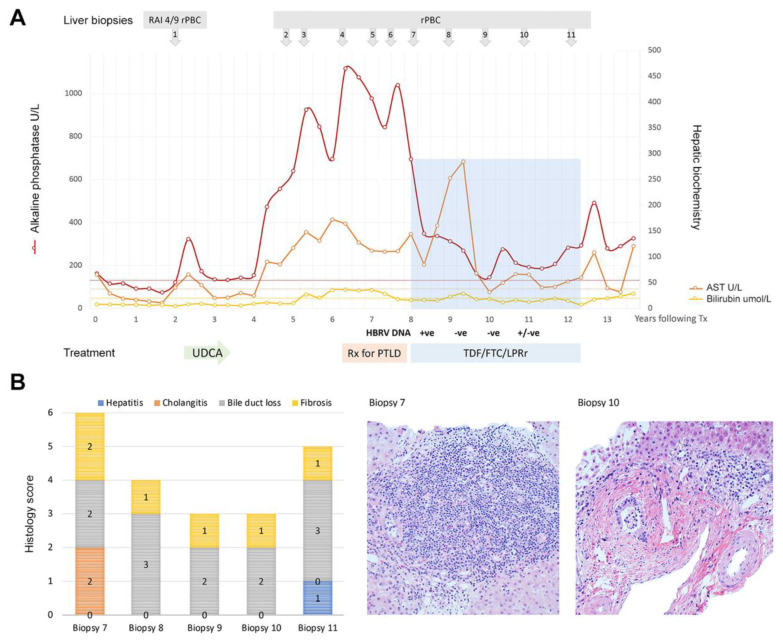
Attenuation of cholangitis by combination ART to treat recurrent PBC following liver transplantation. (**A**) A 21-year-old female developed histological and biochemical evidence of recurrent PBC (rPBC) with mild rejection (RAI: rejection activity index 4/9) within 3 years of liver transplantation. This was treated with ursodeoxycholic acid (UDCA) to accompany her immunosuppression of tacrolimus and mycophenolic acid. Following partial biochemical resolution, her cholangitis worsened, and she developed EBV viremia, mandating a reduction in immunosuppression and use of valganciclovir. She developed post-transplantation lymphoproliferative disorder (PTLD) with an aggressive diffuse large B-cell lymphoma. She was treated with methotrexate, Ara-C, hydrocortisone, and Rituximab, which resulted in severe biochemical cholangitis. She was found to be positive for HBRV DNA in whole blood and following discussions with the patient and liver transplant committee, the patient commenced combination ART with FTC/TDF 200/300 mg and LPV/r 800/200 mg (shaded in blue). (**B**) Prior to the commencement of cART, there was histological evidence of progressive disease with fibrosis, ductopenia and cholangitis (biopsy 7), and cART was associated with a marked reduction in cholangitis and interface hepatitis (biopsy 10). She experienced a marked reduction in alkaline phosphatase with cART, but this was also associated with marked biochemical hepatitis and inhibition of tacrolimus metabolism. Four years after commencing antiviral treatment (year 12), the patient found it difficult to swallow large tablets because of sensory neuropathy and discontinued cART therapy. More recently, the patient commenced a new regimen with TDF/FTC/RAL with a similar biochemical response. (The serial biopsies are numbered and were evaluated for hepatitis, cholangitis, ductopenia and fibrosis using a modified Nakanuma score [9]; liver biopsies were stained with hematoxylin and eosin and are shown with a magnification 400×. Adapted from [8] with permission.)

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
