# Peer review of "Apples to Apples? A Comparison of Real-World Tolerability of Antiretrovirals in Patients with Human Immunodeficiency Virus Infection and Patients with Primary Biliary Cholangitis"

_viruses, 2022, doi:10.3390/v14030516_

Round 1
Reviewer 1 Report
Topic is relevant. It is time that the scientific community take care of the HBRV and of diseases related to it, both inflammatory and neoplastic. The paper is well written and concise and comes out at the right moment. Honestly, I have no relevant change to suggest.
Author Response
We thank the reviewer for their helpful comments.
Reviewer 2 Report
The manuscript entitled “Apples to apples? A comparison of real-world tolerability of antiretrovirals in patients with Human Immunodeficiency Virus Infection and patients with Primary Biliary Cholangitis” by Turvey et al. is an extensive overview on the topic of the repurposing of combination antiretroviral therapy (cART) for the management of primary biliary cholangitis (PBC) linked to a human betaretrovirus (HBRV) infection. The review is well written and well structured, and gives the reader a comprehensive picture and timely update concerning this topic. Moreover it is relevant and interesting that cART has shown clinical improvement in patients with PBC suggesting that further studies using the newer and better tolerated antiretroviral agents may be warranted. I would just like to report a typo: references number 24 and 26 are repeated in the reference list, therefore the numbering in the text and the reference list should please be revised. After this revision I would suggest that the manuscript is suitable for publication in Viruses.
Author Response
We thank the reviewer for their helpful comments and have resubmitted the manuscript with the required changes.
Reviewer 3 Report
Betaretroviruses including MMTV is a family of viral sequences including HML-2, also known as human endogenous retrovirus K (HERV-K). It is clear that this group has been working on human betaretrovirus (HBRV) for a long time and published many fundamental insights on this topic. However, there is a gap in knowledge between what is widely studied, HMLs or HERV-Ks, and what the authors refer to as HBRV. In order for this review to have broader interest for virologists, Section 1.2 needs to be greatly expanded to review prior literature on HBRV in the context of other endogenous viral sequences and studies by other groups, besides your own, on this topic.
There are major concerns with the figures provided in this review. It’s not appropriate to repurpose an already published data as an adapted figure in a review article. In addition, the figures are poorly made and do not meet the bar for publication. The panels are not labeled, one is missing a y-axis label, there are no rulers on the H&E section, and the acronyms are not defined. Figure 2 y-axis is not acceptable. It should start at 0 and the axis should not be cut more than once. This data cannot be interpreted with such large error bars. Each patient data needs to be plotted. If the authors wish to describe their previously published work, they should simply cite the original work and make a model figure in the review.
The use of ART to potentially treat PBC assumes that HBRV is replication competent and infectious. The authors review their prior work on HBRV expression in PBC patients, but it was not clear whether they had found that these are replication competent. The use of AZT to inhibit MMTV, a replication competent virus, is not analogous to inhibit HBRV in humans. The authors mentioned that the HBRV DNA was lower with ART treatment but did not mention how this affected levels of potentially replicating virus. This question ties back into the broader question of what is HBRV, and besides their own work, what have others reported on this?
Author Response
We thank Reviewer 3 for their helpful comments. We have attached a response to each critique.

Round 2
Reviewer 3 Report
Thank you for addressing the comments thoroughly. I agree that the review has significantly improved in quality. Given that ART has now been shown to target other retroviral elements, I still think the conclusion that ART treatment in HIV- patients leads to reduction of HBRV specifically, and nothing else, needs to be carefully evaluated. However, this is beyond the scope of this review, and I look forward to future work in this area.